# Improving the Experience of Providing Care in Community-Based Pharmacies

**DOI:** 10.3390/pharmacy10040067

**Published:** 2022-06-22

**Authors:** Jon C. Schommer, SuHak Lee, Caroline A. Gaither, Nancy A. Alvarez, April M. Shaughnessy

**Affiliations:** 1College of Pharmacy, University of Minnesota, 308 Harvard Street SE, Minneapolis, MN 55455, USA; leex6829@umn.edu (S.L.); cgaither@umn.edu (C.A.G.); 2R. Ken Coit College of Pharmacy—Phoenix, University of Arizona, 650 East Van Buren Street, Phoenix, AZ 85004, USA; nalvarez@pharmacy.arizona.edu; 3American Pharmacist Association, 2215 Constitution Avenue NW, Washington, DC 20037, USA; AShaughnessy@aphanet.org

**Keywords:** personnel, quality, patient safety, work environments, human factors, ergonomics, stress, wellbeing

## Abstract

This study applied a human factors and ergonomics approach to describe community-based pharmacy personnel perspectives regarding how work environment characteristics affect the ability to perform the duties necessary for optimal patient care and how contributors to stress affect the ability to ensure patient safety. Data were obtained from the 2021 APhA/NASPA National State-Based Pharmacy Workplace Survey, launched in the United States in April 2021. Promotion of the online survey to pharmacists and pharmacy technicians was accomplished through social media, email, and online periodicals. Responses continued to be received through the end of 2021. A data file containing 6973 responses was downloaded on 7 January 2022 for analysis. Qualitative thematic analysis was applied for developing operational definitions and coding guidelines for content analysis of the data. The patterns of responses for the dependent variables were compared among community-based practice setting types (chain, supermarket/mass merchandiser, and independent) and work positions (manager, staff pharmacist, technician/clerk, and owner). Chi-square analysis was used for determining statistically significant differences. The findings showed that personnel working in community-based pharmacies reported undesirable work environments and work stress that affected their ability to perform assigned duties for optimal patient care and ensure patient safety. Four work system elements were identified that were both facilitators and barriers to the ability to perform duties and ensure patient safety: (1) people, (2) tasks, (3) technology/tools, and (4) organizational context. Acknowledging local contexts of workplaces, giving adequate control, applying adaptive thinking, enhancing connectivity, building on existing mechanisms, and dynamic continuous learning are key elements for applying the HFE (human factors ergonomics) approach to improving the experience of providing care in community-based pharmacies.

## 1. Introduction

### 1.1. The Next Generation of Community-Based Pharmacy

Community-based pharmacy practice has been shifting from the traditional “locational convenience” retail strategy to one in which pharmacies are “being organized by their capacity to operate as healthcare access points that provide and are reimbursed for patient care and public health services” [1,2,3,4,5]. Community-based pharmacy practices affiliate with clinics and medical centers as part of comprehensive integrated care models. In addition, they are part of vertical integration strategies with insurance companies, wholesalers, manufacturers, integrated delivery networks, pharmacy benefit management companies, pharmacies, clinics, and medical centers [2,6,7]. These shifts are consistent with the triple aim for healthcare that strives to (1) improve the individual experience of care, (2) improve the health of populations, and (3) reduce the cost of healthcare [8].

From an organization-level perspective, these transformations can help (1) provide coordinated services at lower cost, (2) improve access to services, (3) leverage data, and (4) bear financial risk for the health outcomes of patient populations [2,6,7]. Healthcare consumers typically are in favor of extended pharmacist services [9,10,11] and would be willing to accept them [12]. Innovations are being designed for positioning pharmacists to provide a comprehensive array of services directly to consumers [13,14,15,16,17,18,19,20,21,22,23,24,25,26,27,28]. Baines and colleagues [29] described these transformations as a “blended pharmacy practice” work system and process design. New ways of delivering products, managing inventory, and reimbursing for product cost are being developed. At the same time, new ways for recruiting and connecting patients with practitioners, achieving patient outcomes, organizing space for patients to receive services, and being reimbursed for value-based outcomes are emerging [29].

### 1.2. The Need for Improving the Experience of Providing Care

These transformations in community-based pharmacies—and in health systems overall—create uncertainty and stress for personnel. Sikka, Morath, and Leape [30] pointed out that successful achievement for innovative change is an “engaged and productive workforce” that finds meaning and joy in its work. Meaning refers to a sense of importance of daily work. By joy, they refer to the feeling of success and fulfillment that results from meaningful work [30]. In this context, they proposed a fourth aim for healthcare that would also include “improving the experience of providing care” [8,30]. Sikka, Morath, and Leape argued that the following:


*Complex, intimate caregiving relationships have been reduced to a series of transactional demanding tasks, with a focus on productivity and efficiency, fueled by the pressures of decreasing reimbursement. These forces have led to an environment with lack of teamwork, disrespect between colleagues, and lack of workforce engagement.*


They proposed that such dysfunction in healthcare is a byproduct of its shift from a public service to a business-driven model during the latter half of the 20th century [30]. The restoration of meaning and joy for the healthcare workforce depends on physical and psychological freedom from harm, neglect, and disrespect [30]. The challenge is how to accomplish this in evolving healthcare systems.

### 1.3. The Experience of Providing Care in Community-Based Pharmacies

Community-based pharmacy personnel have been experiencing the dysfunction at their work that Sikka, Morath, and Leape described. The 2019 National Pharmacist Workforce Survey found that 71% of pharmacists rated their workload as “high or excessively high”, and job satisfaction was at the lowest point in 20 years [31]. Furthermore, 69% of pharmacists who reported working full-time in 2019 reported that their workload “increased” or “greatly increased” compared to 1 year ago. Across practice settings, both the highest and the lowest proportions of pharmacists rating their workload as “high” or “excessively high” were in community-based pharmacies. The two highest were in chain (91%) and mass merchandiser (88%) pharmacy settings, and the lowest proportion reported across all categories was in independent community (48%) pharmacy settings [31] (but still almost half of that group).

The onset of the COVID-19 global pandemic stretched the healthcare workforce further, including pharmacists, to a breaking point [32,33]. Community-based pharmacists not only provided safe access to medications during periods of quarantine, but also provided COVID-19 testing, vaccinations, treatments, and supplies such as masks and sanitizers during the pandemic. Pharmacy personnel’s overwhelming workload has been linked to patient safety concerns regarding medication errors [34], and student pharmacists’ once positive views of the profession of pharmacy have declined [35].

During 2021, the American Pharmacists Association (APhA) and National Alliance of State Pharmacy Associations (NASPA) sponsored a survey of pharmacy personnel in the United States [36]. The findings showed that pharmacy workplaces were so stressful in 2021 that personnel were unable to meet both clinical and nonclinical duties. The stressful conditions contributing to employee burnout and personnel were at a breaking point where adjustments to team training, roles, and responsibilities were not able to be made quickly enough to adapt to change and meet all of their duties. Time allocation, workflow, staffing, policies, payment, and patient expectations/demands were identified as contributors to workplace situations that can increase the risk of medication errors or near misses. Thus, stressful conditions created threats to patient safety as well [36].

### 1.4. Application of a Human Factors and Ergonomics (HFE) Approach

Most of the factors of concern identified in the APhA/NASPA study relate to work systems and processes of care, which are under the direct control of the employer and management [36]. Thus, there are opportunities to address issues in an expedient manner using human factors and ergonomics (HFE) approaches [37]. These approaches can help guide adjusting quality improvement of work systems and care processes. Carayon and Perry [37] suggested that acknowledging local contexts of workplaces, giving adequate control, applying adaptive thinking, enhancing connectivity, building on existing mechanisms, and dynamic continuous learning are key elements for applying the HFE approach to quality improvement. On the basis of the HFE approach, Carayon and colleagues [37] developed the Systems Engineering Initiative for Patient Safety (SEIPS) model of work system and patient safety for describing various work system barriers and facilitators within the experience of providing care. The model includes five work system elements: (1) the people (at the center of the work system), (2) tasks, (3) tools and technology, (4) physical environment, and (5) organizational context [37].

### 1.5. Study Objectives

In light of findings that link challenging experiences for providing care in community-based pharmacies with the ability to perform assigned tasks and ensure patient safety [36], this study applied a human factors and ergonomics approach to address the following **study objectives** for community-based pharmacy work environments:


Assigned Duties


Describe personnel perspectives regarding how work environment characteristics affect the ability to perform the duties necessary for optimal patient care.Identify work system facilitators to the ability to perform the duties necessary for optimal patient care.Identify work system barriers for the ability to perform the duties necessary for optimal patient care.


Patient Safety


4.Describe personnel perspectives regarding how contributors to stress affect the ability to ensure patient safety.5.Identify work system facilitators to the ability to ensure patient safety.6.Identify work system barriers for the ability to ensure patient safety.

## 2. Materials and Methods

### 2.1. Data Source

Data were obtained from the 2021 APhA/NASPA National State-Based Pharmacy Workplace Survey [36], launched nationally in April 2021 by APhA and state pharmacy associations. Promotion of the online survey to pharmacists and pharmacy technicians was accomplished through social media, email, and online periodicals. Responses continued to be received through the end of 2021. A data file containing 6973 anonymous responses was downloaded on 7 January 2022 for analysis.

For the purpose of this study, usable responses were from those who reported that they worked in a community-based pharmacy (chain, supermarket/mass merchandiser, and independent) and reported their work position (manager, staff pharmacist, technician, and owner). Pharmacy type was defined using the work of Olson et al. [1]. Out of the 6973 responses, 4606 (66%) met our inclusion criteria. Table 1 summarizes the distribution of respondent types.

### 2.2. Study Variables

The measures for work environment and contributors to stress were developed by APhA/NASPA Work Group members using the Wellbeing Index for Pharmacy Personnel (https://www.mywellbeingindex.org/versions/pharmacist-well-being-index, accessed on 7 January 2022), the Tennessee Pharmacists Association workplace survey fielded in the first half of 2020, and a report on Pharmacist’s Fundamental Responsibilities and Rights (https://www.pharmacist.com/pharmacistsresponsibilities, accessed on 7 January 2022). 


Work Environment


There were 12 items developed for the survey [36] that focused on the respondent’s work environment and how time allocation, staffing, policies, payment for services, and workflow design affected their ability to meet both clinical and nonclinical duties. The 12 items were as follows:Sufficient time is allocated for me to safely perform administrative/nonclinical duties.Non-pharmacist staff personnel are available for shifts sufficiently to meet clinical duties.Sufficient time is allocated for me to safely perform patient care/clinical duties.Sufficient non-pharmacist staff personnel are available during shifts to meet administrative/nonclinical duties.Employer policies facilitate my ability to safely perform administrative/nonclinical duties.Sufficient pharmacists are available during shifts to meet patient care/clinical duties.Sufficient pharmacists are available during shifts to meet administrative/nonclinical duties.Sufficient pharmacists overlap and procedures exist to ensure transfer of information and status.Payment for pharmacy services supports our ability to meet clinical and nonclinical duties.Employer policies facilitate my ability to safely perform patient care/clinical duties.Workflow design facilitates my ability to meet nonclinical duties.Workflow design facilitates my ability to meet clinical duties.

The scores for each of the 12 items were summed into an overall “work environment index”. Respondents rated each item from 1 = strongly agree to 5 = strongly disagree. Therefore, the range of scores was from 12 to 60 (theoretical midpoint = 36). Higher scores reveal a higher disagreement with the items. For analysis, the proportion of respondents scoring over the theoretical midpoint of 36 were considered to have “undesirable” work environment characteristics.


Contributors to Stress


There were 13 questions developed for the survey [36] that focused on the respondent’s contributors to stress. The items related to time allocation, workflow, staffing, policies, payment, patient expectations/demands, safety, and harassment/bullying. Each item was rated in terms of how likely each situation contributes to medication errors or near misses on a scale from 1 = very likely to 5 = very unlikely. Thus, the term “contributors to stress” in this survey represents respondents’ views relating to how situations in their workplace contribute to the likelihood of making medication errors or near misses in their work. It is an indicator of potential threats to patient safety. The 13 items were as follows:Interruptions from telephone calls.Inadequate staffing.Patient expectations or demands.Inability to practice pharmacy in a patient-focused manner.Inadequately trained pharmacy personnel.Harassment/bullying from patients/customers.Insurance issuesNon-pharmacy managers’ lack of understanding/knowledge of pharmacy practice regulations.Completion of paperwork or reports.Inconsistent enforcement of workplace policies.Lack of workplace safety.Lack of constructive performance feedback.Harassment/bullying from manager or coworkers.

The scores for each of the 13 items were reverse-coded and then summed into an overall “stress index score”. Therefore, the range for the stress index score was from 13 to 65 (theoretical midpoint = 39). Higher scores reveal a higher likelihood of medication errors or near misses. For analysis, the proportion of respondents scoring over the theoretical midpoint of 39 were considered to have “undesirable” stress that contributes to the likelihood of making medication errors or near misses.


Work System Barriers and Facilitators


Two open-ended questions were used for collecting data regarding work system barriers and facilitators regarding the ability to perform the duties necessary for optimal patient care. They were as follows:What factors have positively impacted your ability to perform the duties necessary for optimal patient care for your patients?What factors have negatively impacted your ability to perform the duties necessary for optimal patient care for your patients?

Two open-ended questions were used for collecting data regarding work system barriers and facilitators regarding the ability to ensure patient safety. They were as follows:What factors have positively impacted your ability to ensure patient safety?What factors have negatively impacted your ability to ensure patient safety?

For these four open-ended questions, 8407 written comments were submitted by community-based pharmacy survey respondents.

### 2.3. Thematic Analysis

To determine if the 8407 written comments could be aligned with the five work system elements in the SEIPS model, qualitative thematic analysis was conducted in an inductive and interpretive manner [38,39,40]. The comments were read several times by five investigators (N.A., C.G., S.L., J.S., and A.S.) independently, and the main themes were extracted [40]. Comments referring to a particular theme were grouped [40,41], and the interpretations were discussed among the five researchers who conducted the thematic analysis. The findings revealed that the emergent themes aligned well with the five work system elements in the SEIPS model: (1) the people (at the center of the work system), (2) tasks, (3) tools and technology, (4) physical environment, and (5) organizational context [37]. This affirmed that subsequent content analysis of the written comments could apply these five work system elements as coding categories.

Exemplars for the five work system elements are highlighted next. These were used methodologically for developing operational definitions for content analysis.


People



*If we feel overwhelmed, we make a plan to get everything done. We have daily huddles to ensure good communication and we have tasks assigned to specific people to ensure that work gets done.*



*I work long hours 12–17 h with no meal breaks. I am on my feet that whole time. I have to do multiple things at once in a futile attempt to keep up with the amount of work I am expected to get done. This is a recipe for disaster in terms of medication errors—I am a tired, distracted pharmacist.*



Tasks



*Our clear tasks that include checks and balances with barcode scanning, data entry review, product review, and an aligned computer system are highly effective at catching errors before reaching the patient.*



*Patient care queue calls. I understand the need for them, but when it’s busy, it’s hard to complete them. “Tasks” are constantly being added that feel unattainable. Right now, my store has a goal of 2 “extended vaccinations” per day. It’s extremely hard to get that when most people are trying to receive COVID vaccines. Also, we have outdated equipment. Our computers and scanners are constantly freezing.*



Tools and Technology



*Technology. Quick links to drug databases, computers that crosscheck fill history with current meds to check for interactions.*



*Far too many apps and programs to manage and be judged on. It takes away from the basic workload that truly makes a difference.*



Physical Environment



*Having a separate counseling room to go and shut the door if I need to concentrate. Days I am able to take uninterrupted breaks and refresh my mind so I can focus.*



*No designated room for giving immunizations. Expectation is to use the waiting room area with other customers waiting in line, family members who may have come in with patient. Huge risk of needle sticks.*



Organization



*Board of directors openly solicits input from all personnel regarding new policies and revising old ones.*



*Not understanding the pressure on pharmacy and how understaffed we are and constantly telling us what to do instead of providing us resources to perform them!*


### 2.4. Content Analysis

On the basis of the thematic analysis and relevant literature regarding the HFE approach and SEIPS model [37,42,43], coding guidelines for the five work system elements were reviewed, modified, and operationalized by five researchers (N.A., C.G., S.L., J.S., A.S.) who met in person for this purpose. All investigators agreed upon major themes [44], and operational definitions for coding categories were finalized as follows:
**People**People’s skills, motivation, needs, familiarity, physical and psychological characteristics, roles, staffing levels, commitment, teamwork, communication, and relationships.**Tasks**Job content, job demands, job support, performance pressure, and time pressure.**Tools & Technology**Information and communication technologies, electronic records, work tools, devices, usability, feasibility, fit, data sources, automation, and maintenance.**Physical Environment**Physical space, rooms, windows, barriers, signage, security, lighting, temperature, and locations.**Organizational Context**Organizational culture, leadership, management and policies relating to metrics, goals, autonomy, freedom, oversight, workflow, supply chain, training, onboarding staff, and information/communication overload.

Two researchers (S.L. and J.S.) were trained to conduct coding for a relatively small number of comments to assess inter-judge reliability. The researchers were trained on the rules and procedures for coding, and they independently scored each comment. Inter-judge reliabilities were then calculated for 78 coding decisions using the Perrault and Leigh reliability index (I), as follows:I = {[(F/N) − (1/k)] [k/(k − 1)]}^1/2^,(1)
where F is the observed frequency of agreement between judges, N is the total number of judgments, and k is the number of categories [45]. The inter-judge reliability score was 0.96. In light of a reliability score well above the recommended level of 0.90, coding was completed by one researcher (J.S.).

### 2.5. Research Team and Reflexivity

As thematic and content analyses were completed, team member reflexivity was conducted so that assumptions were acknowledged and documented as part of the research process [40,46]. The research team consisted of five people (N.A., C.G., S.L., J.S., and A.S.). Two members (J.S. and C.G.) have experience in pharmacist workforce and quality of work life research. Two members (N.A. and S.L.) hold PharmD degrees and have experience in advanced clinical care practice. Two members (N.A. and A.S.) are actively engaged in legislative policy and advocacy work. All five team members hold licenses to practice pharmacy. Each member of the team interacts with pharmacists and student pharmacists on a regular basis. Each member of the team has work experience in community-based pharmacy practice, and these experiences were shared during analytic discussions. The background of the research team provides strengths to this project that helped analyze and interpret the data collected. Personal presuppositions were noted and accounted for in how the analysis may have been influenced.

### 2.6. Rigor

Confirmability was supported by reaching congruence among research team members regarding the data’s relevance, sense, and accuracy [40,46]. Credibility and authenticity were reinforced by multiple readings of the text, documentation, and intercoder checks [40]. This helped assure that data interpretation echoed the respondents’ words and not the biases and viewpoints of the research team [40,46]. Transferability [40] was supported by using a large representation of the population of interest to get a broad range of reflections about people’s experiences and views [47,48]. Thematic saturation [49,50] was reached when thick, vivid descriptions were attained.

### 2.7. Descriptive Analysis by Practice Setting and Position

Through the coding, tabulation, and recoding processes described in earlier sections of this paper, the resultant dependent variables were as follows:Work environment: The proportion of respondents scoring over the theoretical midpoint on the work environment index. This represents undesirable work environment characteristics that interfere with the ability to perform necessary duties.Contributors to stress: The proportion of respondents scoring over the theoretical midpoint on the stress index. This represents undesirable stress that contributes to the likelihood of making medication errors or near misses.Facilitators: The proportion of respondents writing a positive comment relating to (1) person, (2) tasks, (3) tools and technology, (4) physical environment, or (5) organizational context.Barriers: The proportion of respondents writing a negative comment relating to (1) person, (2) tasks, (3) tools and technology, (4) physical environment, or (5) organizational context.

The patterns of responses for these dependent variables were compared among community-based practice setting type (chain, supermarket/mass merchandiser, and independent) and work position (manager, staff pharmacist, technician/clerk, and owner). Chi-square analysis was used for determining statistically significant differences (set at *p* < 0.01, in light of relatively large sample sizes for some cells in the analysis).

## 3. Results

The findings are presented in two sections. The first section focuses on the ability to perform assigned duties, and the second section focuses on the ability to ensure patient safety. 

### 3.1. Assigned Duties

Table 2 summarizes the findings for the proportion of respondents with undesirable work environments that affected their ability to perform necessary duties for optimal patient care. Out of the 4606 survey respondents, 3724 (81%) answered this question. The findings revealed that 96% of respondents from chain pharmacies reported undesirable work environments, followed by supermarket/mass merchandiser (90%), and independent (27%). Manager, staff, and technician positions were similar in their patterns of responses within each pharmacy type. For independent pharmacies, the position of “owner” had a significantly lower proportion than the other position types.

Table 3 summarizes the findings for the proportion of respondents who reported each of the five work system elements that affected their ability to perform assigned duties for optimal patient care. Out of the 4606 survey respondents, 1273 (28%) wrote a positive comment, and 2555 (55%) wrote a negative comment. Chi-square analysis showed that the pattern of responses did not differ significantly by position (manager, staff, technician, and owner). Therefore, findings are reported by pharmacy type only.

For positive comments, the most commonly reported work system elements related to organizational context (34%), people (31%), and technology/tools (23%). Respondents from independent pharmacies were more likely to report organizational context and people and less likely to report technology/tools than the other two practice types.

For negative comments, the most commonly reported work system elements related to people (39%), tasks (30%), and organizational context (29%). Respondents from independent pharmacies were more likely to report organizational context (53%) than chain (28%) and supermarket/mass merchandiser (25%) practice types. Independent pharmacies were less likely to report tasks (9%) than the other two practice types (32% and 29%, respectively).

### 3.2. Patient Safety

Table 4 summarizes the findings for the proportion of respondents with undesirable contributors to stress that affected their ability to ensure patient safety. Out of the 4606 survey respondents, 3233 (70%) answered this question. The findings revealed that 95% of respondents from chain pharmacies reported undesirable work environments, followed by supermarket/mass merchandiser (89%), and independent (58%). Manager, staff, and technician positions were similar in their patterns of responses for chain and supermarket/mass merchandiser pharmacy types. For independent pharmacies, the positions of “technician” and “owner” were significantly lower than the other position types.

Table 5 summarizes the findings for the proportion of respondents who reported each of the five work system elements that affected their ability to ensure patient safety. Out of the 4606 survey respondents, 1868 (41%) wrote a positive comment, and 2710 (59%) wrote a negative comment. Chi-square analysis showed that the pattern of responses did not differ significantly by position (manager, staff, technician, and owner). Therefore, findings are reported by pharmacy type only.

For positive comments, the most commonly reported work system elements related to people (32%), technology/tools (24%), organizational context (22%), and tasks (21%). Respondents from independent pharmacies were more likely to report people (37%) and organizational context (27%) and less likely to report technology/tools (19%) and tasks (15%) than the other two practice types.

For negative comments, the most commonly reported work system elements related to tasks (37%), people (36%), and organizational context (22%). Respondents from independent pharmacies were more likely to report organizational context (44%) than chain (19%) and supermarket/mass merchandiser (22%) practice types. Independent pharmacies were less likely to report tasks (22%) than the other two practice types (39% and 39%, respectively).

## 4. Discussion

### 4.1. Limitations

Before the findings are discussed, several limitations should be considered. The results did not use a random sample of pharmacy personnel. Thus, the findings should be used for gaining insight and not for making estimates for or generalizing to the entire population of pharmacy personnel. Not all survey respondents provided written comments. It is likely that those who wrote comments had strong opinions or were interested in the topic. A human factors and ergonomics approach was used for developing the coding in content analysis. Specifically, five work system elements from the Systems Engineering Initiative for Patient Safety (SEIPS) model were used for framing the coding process. If another framework or set of definitions is used, the findings could be different.

### 4.2. Undesirable Work Environments and Work Stress

Collectively, about nine out of every 10 respondents working in chain, supermarket, and mass merchandiser community pharmacies reported undesirable work environments and work stress that affected their ability to perform assigned duties for optimal patient care and ensure patient safety (Table 2 and Table 4). The pattern of responses among managers, staff pharmacists, and technicians was similar. It appears that these practice settings were strained during 2021 as workload and tasks expanded to unsustainable levels. Although all position types reportedly sensed the overwhelming stress, training and staffing were not able to keep up with these levels of workload and added tasks. In contrast, respondents working in independent pharmacies reported undesirable work environments (27%) and work stress (58%) at relatively lower levels. For each, owners reported lower levels that other position types. While still a challenge, it appears that independent pharmacies were able to adapt relatively quickly with their smaller, more nimble organizational structure.

### 4.3. Application of Human Factors and Ergonomics for Improving the Experience of Providing Care

The findings showed that four out of the five work system elements in the SEIPS model were described by respondents as both barriers and facilitators to the ability to perform duties and ensure patient safety. The only work system element that was not mentioned very often was physical environment. This shows that expensive modifications to physical structures may not be as necessary for improving the experience of providing care in community-based pharmacies as are people, tasks, technology/tools, and organizational context.

For independent pharmacies, “organizational context” emerged as a key element. Comments from these respondents often described challenges with supply chain and reimbursement issues that adversely impacted their organization. Thus, while independent pharmacies might be able to adapt to new tasks and staffing strategies, imbalances in power dependence structures for their contracting with other organizations are creating challenges. In contrast, the other pharmacy practice types are large corporations that have market power for negotiating contracts, investing in technology/tools, and applying corporate-level metrics for meeting challenges. Such approaches appear to help promote efficiencies to meet contract requirements, but might decrease flexibility and autonomy for staffing and training needs to accomplish new tasks at local pharmacy levels. To this point, Carayon and Perry [37] suggested that acknowledging local contexts of workplaces, giving adequate control, applying adaptive thinking, enhancing connectivity, building on existing mechanisms, and dynamic continuous learning are key elements for applying the HFE (human factors ergonomics) approach to quality improvement. Rigid, top-down management is not the best fit for accomplishing these goals. As Sikka, Morath, and Leape pointed out, healthcare involves complex, intimate caregiving relationships [30]. Pharmacy practice requires professional judgement, clinical decision making, patient-centered tailoring of medication action plans, and team-based collaboration. Within such complex, intimate caregiving relationships, pharmacist practitioners often need to suspend routine work processes in order to address serious patient needs. Designing community-based pharmacy as a mechanized assembly line, directed by business metrics and/or non-pharmacist control, is unwise and has contributed to the decline in the experience of providing care in today’s healthcare system. Furthermore, there is evidence to suggest that this decline has reached a point that is threatening patient safety.

### 4.4. The Future of Improving the Experience of Providing Care in Community-Based Pharmacies

The findings support the application of a human factors and ergonomics (HFE) systems approach for improving the experience of providing care in community-based pharmacies. Even before the COVID-19 pandemic, transformations in community-based pharmacies were taking place in order to increase productivity and efficiency to address the challenge of decreasing reimbursement. In 2019, the majority of community-based pharmacists reported high or excessively high workloads, low job satisfaction, increasing demands, and stress [31]. The onset of the pandemic exposed and amplified these existing challenging work conditions and raised patient safety concerns.

These challenges will continue and evolve. We propose that healthcare systems will need to respond to continued external and internal pressures [51,52]. Halfon and colleagues described the need for change from current “coordinated healthcare systems” to new “community-integrated healthcare systems” [53]. Rather than a focus on services, quality outcomes, costs, and provider networks, the next healthcare systems will expand to include population and community health outcomes and optimizing health over people’s life spans and across generations [53]. This would involve integration of healthcare networks into community organizations and include psychosocial services, wellness care, and long-term time horizons.

Community-based pharmacy is already entering this new domain and is likely to be central for the success of community-integrated healthcare systems. More transformation is needed, including updated funding incentives, and this will again create uncertainty and stress for personnel. It will be more important than ever to improve the experience of pharmacist-delivered care so that community-based pharmacies can fulfill their responsibilities that they are ready to meet in the emerging community-integrated healthcare systems. Meaning and joy for personnel depend on physical and psychological freedom from harm, neglect, and disrespect [30]. We advise that rigid, metric-laden approaches would again frustrate the experience of providing care in community-based pharmacies.

## 5. Conclusions

The findings showed that personnel working in community-based pharmacies reported undesirable work environments and work stress that affected their ability to perform assigned duties for optimal patient care and ensure patient safety. It appears that the onset of the COVID-19 pandemic exposed and amplified these challenges. Smaller, more nimble organizations were more able to adapt to new tasks and staffing strategies needed during the pandemic. In contrast, larger corporations were able to use market power to negotiate contracts, invest in technology, and apply corporate-level metrics for meeting efficiency challenges. The findings suggest, however, that top-down, rigid management interfered with pharmacy personnel’s experience of providing care, their ability to exercise professional judgement, patient-centered tailoring of medication action plans, patient safety, and team-based collaboration.

More attention must be paid to improving the experience of providing care in community-based pharmacies. The human factors and ergonomics (HFE) approach using the components of SEIPS can address these issues [37,42,43]. On the basis of the principles described in these domains, we propose that acknowledging local contexts of workplaces, giving adequate control, applying adaptive thinking, enhancing connectivity, building on existing mechanisms, and dynamic continuous learning will help address these challenges.

## Figures and Tables

**Table 1 pharmacy-10-00067-t001:** Number of respondents by pharmacy type and work position.

	Chain	Supermarket/Mass Merchandiser	Independent	Total
**Manager**	1092	337	158	1587
**Staff Pharmacist**	1455	547	245	2247
**Technician**	413	119	37	569
**Owner**	0	0	203	203
**Total**	2960	1003	643	4606

**Table 2 pharmacy-10-00067-t002:** Proportion of respondents with undesirable work environments * that affected their ability to perform assigned duties for optimal patient care.

	Chain (*n* = 2433)	Supermarket/Mass Merchandiser (*n* = 833)	Independent (*n* = 458)	Overall (*n* = 3724)
**Manager (*n* = 1311)**	96%	93%	39%	90%
**Staff pharmacist (*n* = 1836)**	97%	89%	31%	89%
**Technician (*n* = 438)**	95%	86%	30%	90%
**Owner (*n* = 139)**	-	-	12%	12%
**Overall (*n* = 3724)**	96%	90%	27%	87%

Chi-square *p*-value <0.001 for overall associations in the table. * **Undesirable work environment** = proportion of respondents scoring over the theoretical midpoint of 36 on the work environment index.

**Table 3 pharmacy-10-00067-t003:** Proportion of respondents who reported each of the five work system elements that affected their ability to perform assigned duties for optimal patient care.

	Chain (*n* = 2960)	Supermarket/Mass Merchandiser (*n* = 1003)	Independent (*n* = 643)	Overall (*n* = 4606)
**Positive comments**	(*n* = 710)	(*n* = 319)	(*n* = 244)	(*n* = 1273)
Organizational context	33%	29%	43%	34%
People	29%	30%	42%	31%
Technology and tools	26%	30%	7%	23%
Tasks	10%	9%	7%	9%
Physical environment	2%	2%	1%	2%
**Negative comments**	(*n* = 1835)	(*n* = 564)	(*n* = 157)	(*n* = 2556)
People	38%	45%	34%	39%
Tasks	32%	29%	9%	30%
Organizational context	28%	25%	53%	29%
Technology and tools	2%	1%	2%	2%
Physical environment	<1%	<1%	2%	<1%

Chi-square *p*-value < 0.001 for overall associations in the table. **People** = people’s skills, motivation, needs, familiarity, physical and psychological characteristics, roles, staffing levels, commitment, teamwork, communication, and relationships. **Tasks** = job content, job demands, job support, performance pressure, and time pressure. **Tools and technology** = information and communication technologies, electronic records, work tools, devices, usability, feasibility, fit, data sources, automation, and maintenance. **Physical environment** = physical space, rooms, windows, barriers, signage, security, lighting, temperature, and locations. **Organizational context** = organizational culture, leadership, management and policies relating to metrics, goals, autonomy, freedom, oversight, workflow, supply chain, training, onboarding staff, and information/communication overload.

**Table 4 pharmacy-10-00067-t004:** Proportion of respondents with undesirable contributors to stress * that affected their ability to ensure patient safety.

	Chain (*n* = 2217)	Supermarket/Mass Merchandiser (*n* = 741)	Independent (*n* = 275)	Overall (*n* = 3233)
**Manager (*n* = 1160)**	94%	88%	65%	91%
**Staff pharmacist (*n* = 1627)**	95%	90%	61%	92%
**Technician (*n* = 378)**	94%	86%	53%	91%
**Owner (*n* = 68)**	-	-	44%	44%
**Overall (*n* = 3233)**	95%	89%	58%	90%

Chi-square *p*-value < 0.001 for overall associations in the table. * **Undesirable contributors to stress** = proportion of respondents scoring over the theoretical midpoint of 39 on the stress index.

**Table 5 pharmacy-10-00067-t005:** Proportion of respondents who reported each of the five work system elements that affected their ability to ensure patient safety.

	Chain (*n* = 2960)	Supermarket/Mass Merchandiser (*n* = 1003)	Independent (*n* = 643)	Overall (*n* = 4606)
**Positive comments**	(*n* = 1162)	(*n* = 450)	(*n* = 256)	(*n* = 1868)
People	32%	29%	37%	32%
Technology and tools	24%	26%	19%	24%
Organizational context	21%	21%	27%	22%
Tasks	22%	22%	15%	21%
Physical environment	2%	1%	1%	1%
**Negative comments**	(*n* = 1864)	(*n* = 587)	(*n* = 259)	(*n* = 2710)
Tasks	39%	39%	22%	37%
People	37%	34%	30%	36%
Organizational context	19%	22%	44%	22%
Technology and tools	4%	3%	3%	4%
Physical environment	1%	2%	1%	1%

Chi-square *p*-value < 0.001 for overall associations in the table. **People** = people’s skills, motivation, needs, familiarity, physical and psychological characteristics, roles, staffing levels, commitment, teamwork, communication, and relationships. **Tasks** = job content, job demands, job support, performance pressure, and time pressure. **Tools and technology** = information and communication technologies, electronic records, work tools, devices, usability, feasibility, fit, data sources, automation, and maintenance. **Physical environment** = physical space, rooms, windows, barriers, signage, security, lighting, temperature, and locations. **Organizational context** = organizational culture, leadership, management and policies relating to metrics, goals, autonomy, freedom, oversight, workflow, supply chain, training, onboarding staff, and information/communication overload.

## Data Availability

Data used for this study are stored in electronic format and may be obtained from the corresponding author at schom010@umn.edu.

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
