# Peer review of "Improving the Experience of Providing Care in Community-Based Pharmacies"

_pharmacy, 2022, doi:10.3390/pharmacy10040067_

Round 1

Reviewer 1 Report

Using a human factors and ergonomics approach, the researchers analyze how the pharmacy staff's work environment affects their ability to perform the tasks necessary to provide adequate patient care and how stressors affect the staff's ability to ensure patient safety.

Researchers provide a good introduction to the research topic, contextualize the topic, identify the knowledge gap, and clearly present the research development.

Both the instrument (survey) and the methodology are clear, which would allow other researchers to standardize the study. The results obtained are very good, and put this sector of public health on alert, which evidently needs to be intervened urgently. 

The conclusions are well supported, and although they present alarming results, they are correct. 

It is recommended that the manuscript be accepted for publication in its present form.    

Reviewer 2 Report

It has been an easy-to-read manuscript, which sometimes speaks highly of its authors.

To improve the manuscript, I would change the keywords, not including the one that has already been used in the title. Having different words gains strength in search engines.

The authors make a description based on the proposed questions.

In light of these, I believe that the conclusions should be redrafted so that they are clearer.
